# Anomalously warm weather and acute care visits in patients with multiple sclerosis: A retrospective study of privately insured individuals in the US

Holly Elser[1]*, Robbie M. Parks[2,3], Nuriel Moghavem[4], Mathew V. Kiang[5], Nina Bozinov[4], Victor W. Henderson[4], David H. Rehkopf[6], Joan A. Casey[3]

1 Stanford University School of Medicine, Stanford, California, United States of America, 2 Earth Institute, Columbia University, New York, New York, United States of America, 3 Mailman School of Public Health, Columbia University, New York, New York, United States of America, 4 Department of Neurology and Neurological Sciences, Stanford University School of Medicine, Stanford, California, United States of America, 5 Department of Epidemiology and Population Health, Stanford University, Stanford, California, United States of America, 6 Center for Population Health Sciences, Stanford, California, United States of America

* hollys1@stanford.edu

**Data Availability Statement:** Original, diagnosis-level data tied to individuals, locations, and time are considered personally identifiable health

## Abstract

### Background

As the global climate changes in response to anthropogenic greenhouse gas emissions, weather and temperature are expected to become increasingly variable. Although heat sensitivity is a recognized clinical feature of multiple sclerosis (MS), a chronic demyelinating disorder of the central nervous system, few studies have examined the implications of climate change for patients with this disease.

### Methods and findings

We conducted a retrospective cohort study of individuals with MS ages 18–64 years in a nationwide United States patient-level commercial and Medicare Advantage claims database from 2003 to 2017. We defined anomalously warm weather as any month in which local average temperatures exceeded the long-term average by ≥1.5˚C. We estimated the association between anomalously warm weather and MS-related inpatient, outpatient, and emergency department visits using generalized log-linear models. From 75,395,334 individuals, we identified 106,225 with MS. The majority were women (76.6%) aged 36–55 years (59.0%). Anomalously warm weather was associated with increased risk for emergency department visits (risk ratio [RR] = 1.043, 95% CI: 1.025–1.063) and inpatient visits (RR = 1.032, 95% CI: 1.010–1.054). There was limited evidence of an association between anomalously warm weather and MS-related outpatient visits (RR = 1.010, 95% CI: 1.005–1.015). Estimates were similar for men and women, strongest among older individuals, and exhibited substantial variation by season, region, and climate zone. Limitations of the present study include the absence of key individual-level measures of socioeconomic position (i.e.,

information. These data cannot be shared owing to risks of breaching patient confidentiality. Data are available from the Stanford University Center for Population Health Sciences (contact via phsdatacore@stanford.edu) for researchers who meet the criteria for access to confidential data.

**Funding:** J.A.C. was supported by the National Institutes of Environmental Health Sciences R00 ES027023 and P30 ES009089 (https://www.niehs.nih.gov/); M.V.K. was supported the National Institutes on Drug Abuse K99DA051534 (https://www.drugabuse.gov/); and H.E. was supported by a Stanford University School of Medicine MedScholars research award 32288 (http://med.stanford.edu/medscholars.html). The funders had no role in study design, data collection and analysis, decision to publish, or preparation of the manuscript.

**Competing interests:** The authors have declared that no competing interests exist.

**Abbreviations:** DMT, disease-modifying therapy; ECMWF, European Centre for Medium-Range Weather Forecast; ICD, International Classification of Diseases; MS, multiple sclerosis; RR, risk ratio; STROBE, Strengthening the Reporting of Observational Studies in Epidemiology.

race/ethnicity, occupational status, and housing quality) that may determine where individuals live—and therefore the extent of their exposure to anomalously warm weather—as well as their propensity to seek treatment for neurologic symptoms.

## Conclusions

Our findings suggest that as global temperatures rise, individuals with MS may represent a particularly susceptible subpopulation, a finding with implications for both healthcare providers and systems.

## Author summary

### Why was this study done?

- Heat sensitivity is a common clinical feature of multiple sclerosis (MS), a chronic inflammatory disease of the central nervous system.
- Due to climate change, outdoor temperatures are expected to become more variable.
- Individuals with MS may be particularly at risk from anomalously warm temperatures that deviate from the long-term average to which they are otherwise acclimatized.

### What did the researchers do and find?

- Among more than 100,000 privately insured individuals with MS in the United States, we examined whether MS-related healthcare visits were more frequent during periods of anomalously warm weather.
- During periods of anomalously warm weather, individuals with MS are more likely to present for treatment in an acute care setting (i.e., emergency department or inpatient visits).
- This association persisted in subgroup analyses by sex, was stronger in older age groups, and exhibited differences by season, region, and climate zone.

### What do these findings mean?

- Individuals living with MS may be particularly susceptible to anomalously warm weather, which is expected to occur more frequently as the global climate changes.
- Our results motivate ongoing study of the effects of weather and climate anomalies among medically vulnerable groups, including individuals with MS, in varied geographic settings.

## Introduction

Multiple sclerosis (MS) is a chronic inflammatory disease of the central nervous system, which often follows a relapsing–remitting course. Episodes of inflammation are usually followed by at least partial resolution of symptoms, with gradual accumulation of disability over time [1–4]. The prevalence of MS in the United States is highest of any nation, with an estimated 164.6 cases per 100,000 individuals [5]. MS is more prevalent at higher latitudes globally [6], and recent population studies evidence an increasing female-to-male sex ratio, largely driven by rising incidence in older women [7–11]. Among central nervous system disorders, MS is the second most frequent cause of permanent disability in young adults [12,13].

Heat sensitivity in MS—known as Uthoff's phenomenon—is observed in up to 80% of individuals with MS and occurs when sudden increases in core body temperature lead to temporary recrudescence of prior deficits including weakness, autonomic dysfunction, sensory and visual disturbances, cognitive dysfunction, or fatigue. These symptoms typically develop over several hours and persist until core temperature returns to baseline [14–18]. The underlying mechanism appears related to transient slowing or blocking of neural conduction due to core temperature-induced changes in the excitability of previously demyelinated axons [19–21]. Although these episodes are generally understood to be unrelated to disease progression, they may be difficult to distinguish from true relapse and are frequently associated with substantial morbidity and physiologic impairment [22].

Many health implications of temperature extremes are well documented [23,24]. Morbidity and mortality due to heat exhaustion and heat stroke; exacerbation of underlying cardiorespiratory and other chronic diseases; and suicide and injury-related deaths have all been associated with high temperatures [24–27]. Some health outcomes, such as suicide and injury-related deaths, may result from changes in behavior due to anomalously warm temperatures [28,29]. Certain groups—including older adults, individuals with dementia or underlying mental illness, and those living in poverty or social isolation—are particularly susceptible to heat-related complications [30–32]. These groups may experience health effects related not only to excess heat, but also to less extreme temperatures that nevertheless deviate from the long-term local average. Although heat sensitivity is a recognized feature of MS, research that examines the implications of weather and climate for patients living with this disease remains limited and is generally focused on extreme heat [33]. As anthropogenic greenhouse gas emissions drive global climate change, weather and temperature also are expected to become increasingly variable [25,34,35].

In this retrospective cohort study, we examine whether the risk for MS-related outpatient, emergency department, or inpatient visits is increased during periods of anomalously warm weather. Our analysis leverages a unique administrative database comprised of healthcare claims for over 75 million privately insured individuals in the US between 2003 and 2017. We identify anomalously warm weather at the county level based on the extent to which monthly temperatures deviate from the long-term average temperature. This measure is intended to capture indolent changes in seasonal weather patterns that are not necessarily recognizable as periods of extreme heat, occur year-round, and may have health implications for certain groups, including individuals with MS. We hypothesized that during periods of anomalously warm weather, individuals with MS would be more likely to experience neurological symptoms, resulting from either physiological or behavioral changes, that prompt treatment seeking.

## Methods

Optum's Clinformatics Data Mart Database (OptumInsight, Eden Prairie, Minnesota, US) is a large, de-identified commercial and Medicare Advantage claims database. Patient-level commercial claims data are available from January 1, 2003 to December 31, 2017. Member enrollment data; diagnostic codes from outpatient, emergency department, and inpatient visits; and pharmacy claims are deterministically linked across file types with a unique patient identifier [36]. This study was approved by the Institutional Review Board at Stanford University (IRB-55772). This study is reported as per the Strengthening the Reporting of Observational Studies in Epidemiology (STROBE) guideline for cohort studies (**S1 Strobe Checklist**). An analysis plan was prepared in April 2020, which we have updated to reflect changes made throughout the peer review process (**S1 Analysis Plan**).

### Study design and participants

Individuals eligible for the present study were between the ages of 18 to 64 years, resided in any of the lower 48 states or District of Columbia, were eligible for insurance for at least 1 month, and were identified as having MS based on a previously validated algorithm [37]. This required at least 3 MS-related claims for any combination of inpatient, outpatient, or disease-modifying therapy (DMT) claims within 365 days. MS-related inpatient and outpatient claims were identified using primary diagnostic code 340 from the International Classification of Diseases, Ninth Revision (ICD-9) and G35 from the ICD-10. Prescription claims for DMT included interferon beta-1a-SC, interferon beta-1a-IM, interferon beta-1b-SC, pegylated interferon beta 1b, glatiramer acetate, dimethyl fumarate, fingolimod, siponimod, teriflunomide, cladribine, mitoxantrone, alemtuzumab, ocrelizumab, or natalizumab.

Using insurance eligibility files, we created a person-month dataset in which follow-up extended from the first month in which an individual satisfied our inclusion criteria (i.e., the month in which their third MS-related claim occurred) until the last month of insurance eligibility or the end of the study period on December 31, 2017. In this open cohort, we defined eligible person-months based on insurance eligibility and excluded all person-months of follow-up in which an individual was not eligible for insurance. We made no other restrictions or exclusions based on duration of insurance eligibility.

### Anomalously warm weather

The primary exposure of interest for the present study was anomalously warm weather. In contrast with measures of absolute temperature to which individuals and populations are otherwise adapted, this measure is intended to capture seasonal deviations from long-term weather patterns that are expected to occur more commonly as a consequence of climate change. Temperature data were available through the European Centre for Medium-Range Weather Forecast's (ECMWF) open-access ERA5 reanalysis [38]. Using these data, we calculated the difference between average monthly temperatures and the long-term average temperature for each county, defined as the average each calendar month (January to December) over the study period [28,39–41]. We created an indicator variable that equaled 1 if the average monthly temperature exceeded the long-term county average by at least 1.5˚C and equaled 0 otherwise. We selected this threshold consistent with efforts to limit global temperature rise within this century to below 1.5˚C as described in the Paris Agreement [42].

Alternative parameterizations of our primary exposure included (1) the difference between the monthly average and long-term average temperature defined as a categorical variable; and (2) a series of indicator variables that corresponded to alternative thresholds for anomalously warm weather (when monthly average temperatures were 0.5, 1.0, or 2.0˚C above the long-

term average for each county). Alternative exposure metrics included (1) the average monthly temperature; (2) the number of days per month that exceeded the long-term average for that calendar day by at least 1.5˚C defined as a categorical variable; and (3) an indicator variable for anomalously cool weather, which equaled 1 if average monthly temperatures were at least 1.5˚C below the long-term average.

## MS-related outpatient, emergency department, and inpatient visits

We identified MS-related outpatient, emergency department, and inpatient visits using diagnostic code 340 (ICD-9) and G35 (ICD-10) in the first, second, or third diagnostic position. Because monthly claims exhibited overdispersion due to a point mass at 0 and right skew, we created an indicator variable for each visit type that equaled 1 for any month in which an individual had at least 1 MS-related outpatient, emergency, or inpatient visit—respectively—and equaled 0 otherwise. We created an indicator variable for visits unrelated to MS which equaled 1 for outpatient, emergency, or inpatient visit in which the diagnostic codes 340 and G35 were absent from all diagnostic positions.

## Covariates

Using insurance eligibility files, we determined sex (female or male), age, and calendar year. Using 5-digit ZIP codes from eligibility files, we assigned time-varying county, state, and region of residence (Northeast, South, Midwest, and West) for each respondent. Additional covariates included season (winter, December to February; spring, March to May; summer, June to August; and fall, September to November) and climate zone designations (marine, very cold, cold, mixed-humid, mixed-dry, hot-humid, hot-dry) as defined by the US Department of Energy [43] and assigned based on county of residence.

## Statistical analyses

We examined the association between anomalously warm weather and MS-related healthcare visit in an individual-level repeated measures framework. For all described analyses, the unit of analysis was the person-month. We therefore calculated robust standard errors to account for repeated measures within individuals and potential nonindependence of outcomes within counties. All statistical analyses were performed with R version 3.2.3 (R Foundation for Statistical Computing, Vienna, Austria).

We used generalized linear models with the binomial family and log link to estimate risk ratios (RRs) for the association between anomalously warm weather and MS-related outpatient, emergency department, and inpatient visits, respectively. Models were adjusted for variables identified a priori as potential confounders of the association between anomalies and MS-related healthcare visits. These included sex, continuous age defined using natural splines, and fixed effects for US state and calendar year to account for regional differences in provider networks and secular trends, respectively. We then fitted the above specified log-linear models to estimate associations on the additive scale. Because of sex differences in MS incidence, response to treatment, and patterns of healthcare utilization [8,10,44–46], we repeated our primary analyses separately among men and women.

**Secondary analyses.** We conducted the following a priori specified subgroup analyses. Because MS is characterized by progressive disability [47,48], we repeated our primary analysis within age categories (18 to 25, 26 to 35, 36 to 45, 46 to 55, and 56 to 64 years). Because temperatures vary over space and time, we then examined the association between anomalously warm weather and each type of MS-related visit by season, region, and by season within each

region. Post hoc, we further examined potential heterogeneity in the response to anomalously warm weather within climate zones.

**Sensitivity analyses.** We conducted the following sensitivity analyses. First, we examined the association between the magnitude of positive deviations of monthly average temperatures from the long-term average and MS-related healthcare visits. Second, we examined the association between the number of anomalously warm days per month and MS-related healthcare visits. Third, we repeated our primary analysis using a series of alternative thresholds to define anomalously warm weather (0.5, 1.0, and 2.0°C above the long-term monthly average). These metrics were intended to examine whether deviations of average monthly temperatures of greater magnitude conferred greater risk for MS-related healthcare visits. We further examined the association between average monthly temperature and MS-related visits using natural splines to capture potential nonlinearities. This metric was intended to capture whether high absolute temperatures—rather than deviations—were associated with MS-related healthcare visits.

It is plausible that anomalies during the previous month could increase risk for MS-related healthcare visits, whereas visits should not be related to anomalies that occur in the future. We therefore conducted a sensitivity analysis in which we examined the association between MS-related healthcare visits and anomalously warm weather during the previous month (i.e., 1 month in the past) and during the following month (i.e., 1 month into the future).

Finally, we conducted negative exposure and outcome control analyses as robustness checks [49,50]. We estimated the association between anomalously warm weather and visits unrelated to MS as a negative outcome control analysis. Because numerous health conditions have previously been linked with heat and temperature, no association or an attenuated association between anomalously warm weather and visits unrelated to MS would support an independent relationship between anomalously warm weather and MS-related visits. Because sensitivity to high temperatures—not low temperatures—is a known clinical feature of MS, we conducted a negative exposure control analysis that examined the association between anomalously cool weather and MS-related visits. Finally, we included a negative exposure control analysis in which we created a random permutation of the exposure variable for each individual. In these analyses, a null relationship between temperature anomalies and MS exacerbations provides evidence that the main results are not spurious.

## Results

Of 75,395,344 individuals included in the Optum database, we identified 106,225 (0.1%) with MS. These individuals were followed for a total of 3,785,229 months with an average of 83.3 cases of MS detected per 100,000 person-months of follow-up (**S1 Fig**). MS-related visits were most frequent in Minnesota, Idaho, Alabama, Maine, and Connecticut (**S2 Fig**). Anomalously warm weather occurred most frequently between October and March (**Fig 1**). The study population (**Table 1**) was comprised primarily of women (*N* = 80,212, 76.6%), and the majority of individuals were aged 36 to 55 years at baseline (*N* = 62,643, 59.0%). Approximately 40% of the study population resided in the South (*N* = 52,351), whereas only 3.9% resided in the Northeast (*N* = 3,484). By region, on average, temperatures were highest in the South and lowest in the Northeast with greatest year-round variation observed in the Midwest and the least year-round variation in the West (**S1 Table**).

Anomalously warm weather was associated with increased risk for emergency department visits (RR = 1.043, 95% CI: 1.025 to 1.063) and inpatient visits (RR = 1.032, 95% CI: 1.010 to 1.054). There was limited evidence of an association between anomalously warm weather and MS-related outpatient visits (RR = 1.010, 95% CI: 1.005 to 1.015). Associations were similar for

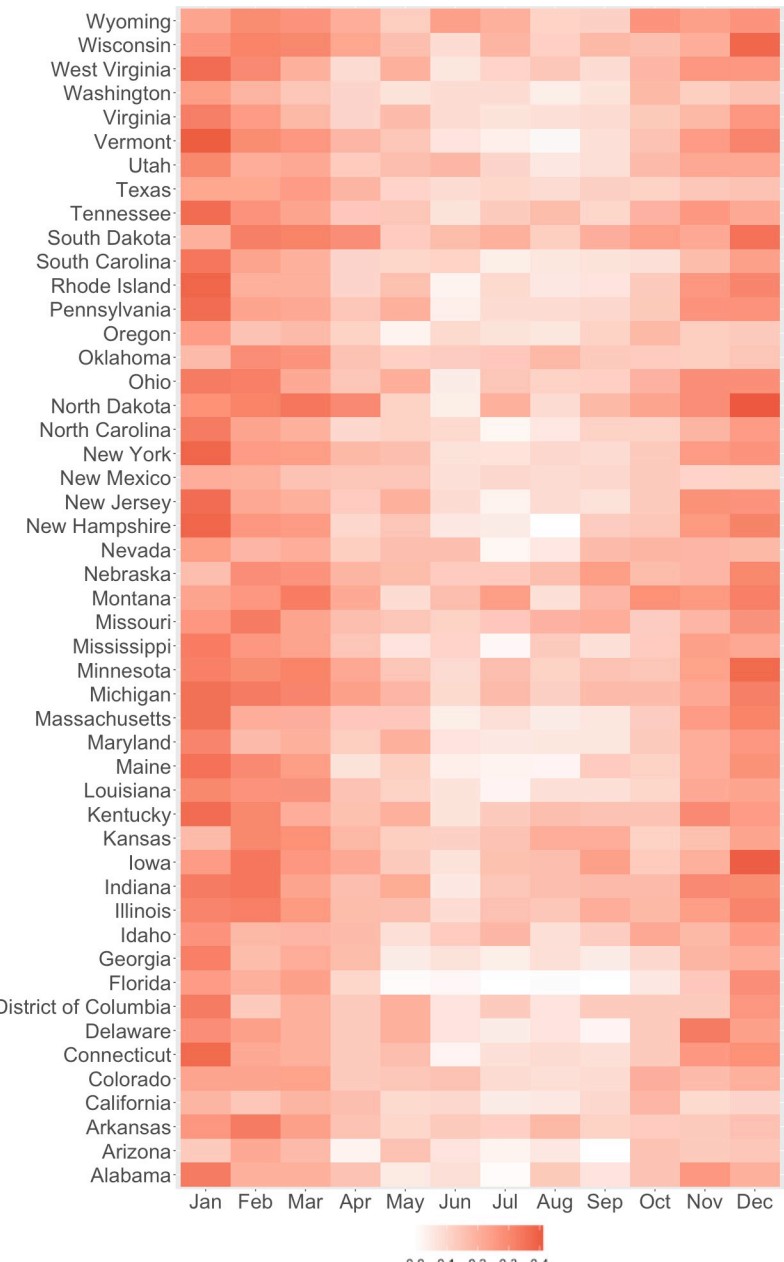

**Fig 1. Frequency of temperature anomalies by month and state, 2003–2017.** We designated anomalously warm weather at the county level as any month in which the average monthly temperature exceeded the long-run average for that month and county by at least 1.5°C. Temperature data were available through the ECMWF's open-access ERA5 reanalysis. Shading intensity corresponds to the number of anomalously warm months divided by the total months of follow-up for each state between January 2003 and December 2017. ECMWF, European Centre for Medium-Range Weather Forecast.

men and women (**Fig 2**, **S2 Table**). Using fitted models from our primary analysis, we estimated that there were an additional 1,960 outpatient (95% CI: 1,126 to 2,794), 592 emergency department (95% CI: 305 to 878), and 1,260 inpatient (95% CI: 892 to 1,628) person-months in which an MS-related visit occurred associated with anomalously warm weather over the course of the study period (**S3 Table**).

**Table 1. Demographic characteristics of the study population (N = 106,225)[1].**

| Demographic characteristics | N (%) |
|---|---|
| **Sex** | |
| Women | 80,212 (75.5) |
| Men | 26,013 (24.5) |
| **Age category, years[2]** | |
| 18–25 | 3,484 (3.3) |
| 26–35 | 15,798 (13.9) |
| 36–45 | 28,567 (26.9) |
| 45–55 | 34,076 (32.1) |
| 56–64 | 23,300 (21.9) |
| **US census region[3]** | |
| Northeast | 4,242 (3.9) |
| South | 52,351 (42.3) |
| Midwest | 28,695 (27.0) |
| West | 20,937 (19.7) |
| **Climate zone[4]** | |
| Very cold | 917 (0.9) |
| Cold | 37,583 (35.4) |
| Mixed-humid | 31,708 (29.8) |
| Mixed-dry | 633 (0.6) |
| Hot-humid | 21,944 (20.7) |
| Hot-dry | 8,883 (8.4) |
| Marine | 4,557 (4.3) |
| **Months of follow-up**–median (IQR)[5] | 24 (11–49) |

[1]Individuals eligible for the present study were aged 18–64 years and resided in any of the lower 48 states including the District of Columbia; were eligible for insurance for at least 1 month; and had at least 3 MS-related diagnostic codes or DMT claims within 365 days.

[2]We defined age at baseline (i.e., at the start of follow-up).

[3]We defined US census regions as the **Northeast** (Connecticut, Massachusetts, Maine, New Hampshire, New Jersey, New York, Pennsylvania, Rhode Island, and Vermont); the **South** (Alabama, Arkansas, Delaware, District of Columbia, Florida, Georgia, Kentucky, Louisiana, Maryland, Mississippi, North Carolina, Oklahoma, South Carolina, Tennessee, Texas, Virginia, and West Virginia); the **Midwest** (Illinois, Indiana, Iowa, Kansas, Michigan, Minnesota, Missouri, Nebraska, North Dakota, Ohio, South Dakota, and Wisconsin); and the **West** (Arizona, California, Colorado, Idaho, Montana, Nevada, New Mexico, Oregon, Washington, and Wyoming).

[4]We used climate zone designations defined by the US Department of Energy Building America Program.

[5]Total months of insurance eligibility from index date (i.e., date of the third MS-related claim) until the individual reached the age of 65, insurance termination, death, or administrative censoring at the end of the study period on December 31, 2017.

DMT, disease-modifying therapy; MS, multiple sclerosis.

## Secondary analyses

By age group, we identified the strongest associations among individuals aged 56 to 64 years for outpatient visits (RR = 1.013, 95% CI: 1.005 to 1.021), emergency department visits (RR = 1.043, 95% CI: 1.015 to 1.073), and inpatient visits (RR = 1.053, 95% CI: 1.031 to 1.073) (**Fig 3**, **S4 Table**).

By season, we observed the strongest associations between warm weather anomalies and MS-related visits in the winter (**S5 Table**). By region, we consistently observed stronger

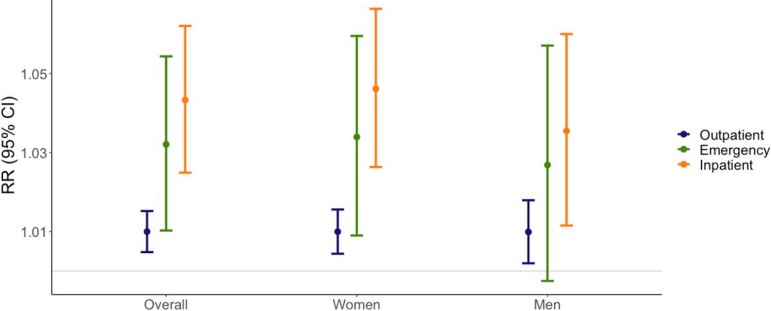

**Fig 2. Association between anomalously warm weather and MS-related outpatient, emergency department, and inpatient visits in the study population overall and by sex, 2003–2017.** We used generalized linear models with the binomial family and log link specified to estimate RRs first in the study population overall and then separately for men and women. We created an indicator variable for anomalously warm weather at the county level which equaled 1 for any month in which the average temperature exceeded the long-run average by 1.5°C and equaled 0 otherwise. For each visit type, the outcome of interest was a monthly indicator variable that equaled 1 if there was at least MS-related outpatient, emergency, or inpatient visit and 0 otherwise. All models were adjusted for continuous age with a natural spline and 3 degrees of freedom and included a set of indicator variables to control for confounding by state of residence and calendar year. The model for the study population overall included a control for categorical sex (male or female). MS, multiple sclerosis; RR, risk ratio.

associations in the South for outpatient visits (RR = 1.024, 95% CI: 1.017 to 1.032), emergency department visits (RR = 1.072, 95% CI: 1.042 to 1.103), and inpatient visits (RR = 1.084, 95% CI: 1.059 to 1.110). Conversely, we observed potentially protective associations in the West, particularly for inpatient visits (RR = 0.960, 95% CI: 0.915 to 1.007) (**S6 Table**). By season and region, we observed the strongest association for emergency department visits in the fall in the Midwest (RR = 1.095, 95% CI: 1.032 to 1.161), in the fall in the Northeast (RR = 1.132, 95% CI: 0.967 to 1.326), in the winter in the South (RR = 1.096, 95% CI: 1.035 to 1.160), and in the summer in the West (RR = 1.076, 95% CI: 0.903 to 1.281) (**S7 Table**). By climate zones, we observed stronger associations in cold and humid climates and weaker or null associations in dry and marine climates. Individuals living in hot-humid climates had the strongest associations for both emergency department visits (RR = 1.109, 95% CI: 1.053 to 1.168) and inpatient visits (RR = 1.090, 95% CI: 1.048 to 1.135). In formal tests of statistical interaction, we

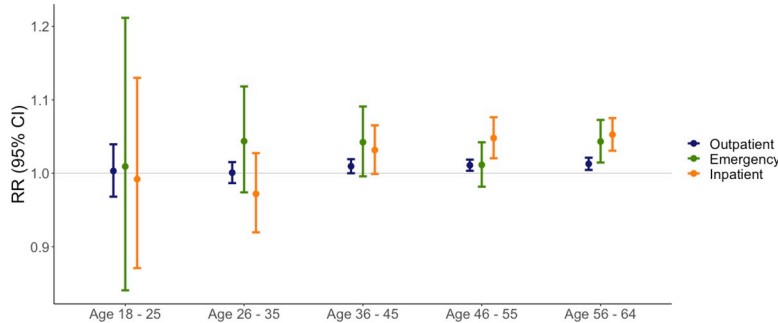

**Fig 3. Association between warm weather anomalies and MS-related outpatient, emergency department, and inpatient visits by age group, 2003–2017.** We used generalized linear models with the binomial family and log link specified to estimate RRs within age groups (18–25; 26–35; 36–45; 46–55; 56–64). We created an indicator variable for anomalously warm weather at the county level which equaled 1 for any month in which the average temperature exceeded the long-run average by 1.5°C and equaled 0 otherwise. For each visit type, the outcome of interest was a monthly indicator variable that equaled 1 if there was at least MS-related outpatient, emergency, or inpatient visit and 0 otherwise. All models were adjusted for sex (male or female) and included a set of indicator variables to control for confounding by state of residence and calendar year. MS, multiple sclerosis; RR, risk ratio.

observed no statistically significant interaction by sex, age group, or season, but found statistically significant interaction by region and climate zone.

## Sensitivity analyses

We found that the greater the magnitude of the difference between monthly average and long-term temperatures, the greater the risk of MS-related emergency and inpatient visits (**S9 Table**). An increase in the number of anomalously warm days per month was also associated with increased risk of MS-related visits, with the strongest associations observed with more than 14 anomalously warm days per month for both emergency department visits (RR = 1.052, 95% CI: 0.990 to 1.117) and inpatient hospital visits (RR = 1.070, 95% CI: 1.024 to 1.117) as compared to months with no anomalously warm days (**S10 Table**). In our analysis of alternative thresholds for anomalously warm weather, associations were weakest with a threshold of 0.5˚C above the long-term monthly average and strongest for the 2.0˚C threshold (**Fig 4, S11 Table**). By contrast, we found no evidence of an exposure–response relationship between average absolute monthly temperature and MS-related visits (**S3 Fig**).

In our analysis with the lagged exposure variable, we found that anomalously warm weather during the previous month was weakly associated with MS-related inpatient visits (RR = 1.028, 95% CI: 1.010 to 1.047). However, we observed no evidence that anomalously warm weather 1 month in the future increased risk for MS-related healthcare visits (**S12 Table**). In our negative outcomes control analysis, the associations between anomalously warm weather and visits unrelated to MS were consistently attenuated as compared with those for MS-related visits (**S13 Table**). In our negative exposure control analyses, we observed no evidence of an association between anomalously cool weather and MS-related visits (**Table 2**) or in our analysis with random permutation of the exposure variable (**Table 3**).

## Discussion

In this retrospective cohort study, we examined whether the risk for MS-related outpatient, emergency department, or inpatient hospital visits was increased during months in which local temperatures exceeded the long-term average by at least 1.5˚C. Our measure of anomalously warm weather was intended to capture more indolent changes in seasonal weather patterns, likely to occur more frequently with climate change, that may not be recognizable as periods of extreme heat, but nevertheless occur year-round and may have health implications for certain groups, including individuals living with MS. Collectively, our results provide preliminary evidence that anomalously warm weather increases risk for MS-related medical visits in the acute care setting.

Despite the widely recognized clinical significance of temperature variation for patients with MS, population research that examines the implications of weather and climate for patients living with this disease remains limited and is generally focused on extreme heat. In one cross-sectional analysis of 40 patients with MS and 40 healthy controls, Leavitt and colleagues found that cognitive status in patients with MS was worse on warmer days [51]. By contrast, Tataru and colleagues found no evidence for increased hospital admissions and relapses in MS patients during the 2003 heat wave in France [33].

We found that during periods of anomalously warm weather, individuals with MS were more likely to present for treatment in an acute care setting in particular (i.e., emergency department or inpatient visits). Sensitivity analyses further suggested that greater deviations in monthly average temperature from the long-term average conferred greater risk for treatment seeking related to MS in the acute care setting. By contrast, we found limited evidence that outpatient visits, which may require a scheduled appointment, were associated with anomalously

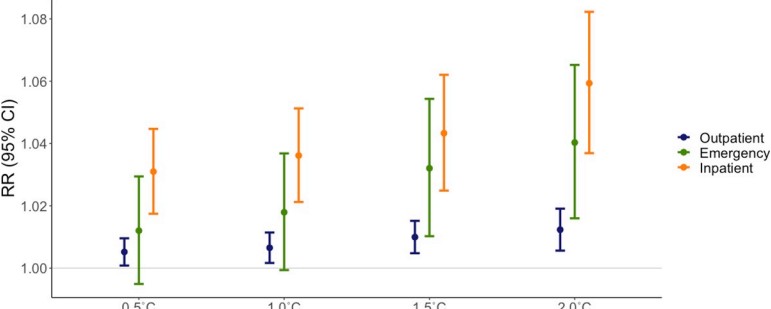

**Fig 4. MS-related outpatient, emergency department, and inpatient visits with alternative thresholds for warm weather anomalies.** We created a range of alternative metrics for anomalously warm weather in which the average temperature exceeded the long-run average for that calendar month and county by 0.5, 1.0, 1.5, or 2.0˚C. We used generalized log-linear models to estimate RRs. For each visit type, the outcome of interest was a monthly indicator variable that equaled 1 if there was at least MS-related outpatient, emergency, or inpatient visit and 0 otherwise. All models were adjusted for continuous age with a natural spline and 3 degrees of freedom, sex (male or female), and included a set of indicator variables to control for confounding by state of residence and calendar year. RR, risk ratio.

warm weather. Although the magnitude of estimated associations was relatively modest, they are consistent with prior studies on the population health effects of environmental exposures [52–54]. In particular, small relative increases in risk may translate to substantial disease burden on the absolute scale when the entire population is exposed.

We found no evidence to support increased risk of MS-related healthcare visits associated with high absolute monthly average temperatures. Because heat sensitivity is a known clinical feature of MS, patients are often counseled to avoid the provocative effects of extreme heat and are therefore more likely to take precautions against high absolute ambient temperatures [22]. It is therefore possible that while individuals with MS are able to successfully avoid exposure to high absolute temperatures they are relatively more susceptible to warmer-than-expected temperatures. In warmer seasons, such as spring or summer, anomalously warm weather could plausibly lead to increases in core body temperature sufficient to increase risk of MS admission via additional heat stress. During anomalously warm winter or fall months, modified behavior,

**Table 2. Negative exposure control: anomalously cool weather and MS-related visits, 2003–2017[1,2,3].**

| | Overall RR (95% CI) | Women RR (95% CI)[2] | Men RR (95% CI) |
|---|---|---|---|
| **Outpatient visits[4]** | 0.987 (0.982–0.992) | 0.987 (0.981–0.993) | 0.987 (0.979–0.995) |
| **Emergency visits** | 1.001 (0.980–1.022) | 0.995 (0.972–1.019) | 1.017 (0.983–1.053) |
| **Inpatient visits** | 1.005 (0.987–1.024) | 1.007 (0.986–1.028) | 1.001 (0.976–1.027) |

[1]We defined anomalously cool weather as any month in which the average temperature was at least 1.5˚C below the long-term average for that month and county.

[2]We defined MS-related visits as those with diagnostic codes 340 (ICD-9) and G35 (ICD-10) for the first, second, or third diagnostic position.

[3]We used generalized linear models with the binomial family and log link specified to estimate RRs. All models included controls categorical sex (male or female), continuous age defined by natural splines with 3 degrees of freedom, and a set of indicator variables for state and calendar year. We calculated robust standard errors to account for potential nonindependence of outcomes within individuals over time and within counties.

[4]Included visits to medical offices, outpatient hospitals, urgent care facilities, independent clinics, walk-in retail health clinics, and state or local public health clinics.

ICD, International Classification of Diseases; MS, multiple sclerosis; RR, risk ratio.

**Table 3. Negative exposure control: random permutation of temperature anomalies, 2003–2017[1,2,3].**

|  | Overall RR (95% CI) | Women RR (95% CI) | Men RR (95% CI) |
|---|---|---|---|
| **Outpatient visits[4]** | 1.002 (0.998–1.006) | 1.004 (0.999–1.009) | 0.996 (0.989–1.004) |
| **Emergency visits** | 1.007 (0.983–1.011) | 1.009 (0.990–1.028) | 1.004 (0.973–1.036) |
| **Inpatient visits** | 0.997 (0.983–1.011) | 1.002 (0.986–1.017) | 0.984 (0.961–1.008) |

[1]We created a negative exposure control variable which represents the random permutation of the order of temperature anomalies for each study participant.

[2]We defined MS-related visits as those with diagnostic codes 340 (ICD-9) and G35 (ICD-10) for the first, second, or third diagnostic position.

[3]We used generalized linear models with the binomial family and log link specified to estimate RRs. All models included controls categorical sex (male or female), continuous age defined by natural splines with 3 degrees of freedom, and a set of indicator variables for state and calendar year. We calculated robust standard errors to account for potential nonindependence of outcomes within individuals over time and within counties.

[4]Included visits to medical offices, outpatient hospitals, urgent care facilities, independent clinics, walk-in retail health clinics, and state or local public health clinics.

ICD, International Classification of Diseases; MS, multiple sclerosis; RR, risk ratio.

such as increased outdoor physical outdoor exertion, could also lead to increases in core body temperature. Collectively, our results therefore suggest that patients with MS may need to work with their healthcare providers to anticipate and protect themselves from warmer-than-expected temperatures even when absolute temperatures are more moderate or even cool.

In subgroup analyses, we observed minimal differences by sex. However, past literature consistently documents significant differences in risk of disease onset [7,8,10], response to treatment [44], disease burden, and health-related quality of life [55,56] in men and women with MS, as well as increased overall healthcare utilization by women compared to men [45,46]. This motivates ongoing examination of potential sex differences in the implications of anomalously warm weather for patients with MS where more direct measures of symptoms, disability, and disease burden are available. In subgroup analysis by age, we found increasingly strong associations between warm weather events and MS-related healthcare visits in older age groups. Age has modified the association between ambient temperature and ED visits for multiple outcomes in prior US-based studies [57,58]. Our finding may reflect the accumulation of neurologic disability with age in patients with MS [59,60], that the health effects of heat are seen more frequently in older adults in general [32,61,62] or some combination thereof.

The associations between warm weather and MS-related hospital visits also demonstrated substantial variation by season, region, and climate zone. By season, we observed the strongest associations during the winter. By region, we observed the strongest associations in the South and the weakest associations in the West, where temperatures exhibited the least year-round variability and where a 1.5˚C deviation from the long-term average therefore reflects the greatest relative deviation. These findings may be partially explained by climate. Consistent with stronger associations during the winter, we also observed some of the strongest associations between anomalously warm weather and MS-related healthcare visits among individuals living in the very cold climate zone. Again, this observation implies behavioral factors [63]. Increased physical exertion, wearing unnecessary layers, or spending time in indoor heated spaces may be important and warrant consideration in future studies. Healthcare visits in very cold climates during warmer months may also reflect limited air conditioning [64]. However, we also observed consistently strong associations in mixed-humid and hot-humid climate zones,

suggesting that both cold and humid climates may modify the relationship between anomalies and MS-related healthcare visits.

Another potential explanation for the observed spatiotemporal heterogeneity in our study are individual-level socioeconomic indicators, which our study data lacked. Even in a population comprised of individuals with private insurance, differences in socioeconomic status may contribute to our pattern of findings as past research consistently demonstrates that lower-income individuals are more susceptible to temperature-related morbidity and mortality [65–68]. This may be due to factors including increased occupational exposure, poor housing quality, poor baseline health, energy insecurity, and the inability to afford full-time air conditioner use during heat events [69]. Future studies should examine how individual-level socioeconomic status modifies the association, with specific consideration to the role of housing conditions and household income whenever possible.

Our findings may also be explained in part by behavior. We found the strongest associations between anomalously warm weather and emergency department visits in the fall in the Northeast and Midwest and in the winter in the South rather than in the summer when absolute temperatures are highest. This pattern may reflect the fact that individuals with MS take fewer precautions against warm temperatures, such as using air conditioning or planning to spend time indoors when average temperatures are typically moderate or cool. A study by Summers and colleagues from Australia found that households that included people with MS spent between 4 and 12 times more on air conditioner use as compared with households without people with MS [70]. Improved energy efficiency and weatherization of residences may represent a cost-effective solution with significant health benefits [71]. Energy vouchers for lower-income individuals living with MS, while a less viable short-term solution, could potentially be used during shoulder seasons when warm weather events appear to pose the greatest risk.

## Limitations

The study population was comprised of individuals covered by a single private insurance provider for at least some time between 2003 and 2017. We aimed to minimize selection bias by making no restrictions in eligibility based on duration or continuity of follow-up. Generalizability is limited because our study does not include individuals 65 years of age or older, the lowest income US residents, or individuals with MS with at least 24 consecutive months of disability who qualified for federal health insurance through Medicare. Nearly half of the study population resided in the South with relatively few individuals from the Northeast included (approximately 4% of the study population). Moreover, given the above-noted regional differences and the unique nature of the US healthcare system, our findings may not generalize to other geographic contexts.

The outcomes of interest in the present study were 3 measures of MS-related healthcare visits (outpatient, emergency department, and inpatient), which are an imprecise proxy for underlying burden of symptoms and disability. Some outcome misclassification is possible if visits related to MS were not coded as such and vice versa. We expect such misclassification to be non-differential and therefore would attenuate our findings. In our negative outcome control analysis, we observe consistently attenuated but persistent associations between anomalously warm weather and MS-related visits. We anticipate that these residual associations reflect some combination of the fact that individuals with MS may have other conditions that increase their sensitivity to temperature anomalies (e.g., cardiovascular disease) and that some of these visits may truly have been related to MS but not coded as such. As expected, we observed no associations for either of our negative exposure control analyses.

As a final limitation, these study data lack a number of important individual-level covariates. Namely, individual-level measures of socioeconomic status such as occupational status, household income, access to air conditioning, housing quality, and race/ethnicity may determine where individuals are located—and therefore the extent of their exposure to anomalously warm weather—and their propensity to seek treatment for neurologic symptoms. Future studies should consider these covariates if available as they are likely important modifiers of the effect of warm weather events on MS-related hospital visits and disease burden and will help inform potential interventions.

## Conclusions

In this retrospective cohort study, we examined the implications of anomalously warm weather for healthcare visits related to MS in a cohort of 106,225 privately insured individuals. We find preliminary evidence that suggests anomalously warm weather—wherein the local temperature exceeded the long-term average to which individuals were otherwise accustomed—increases risk for MS-related emergency department and inpatient hospital visits in particular. Across groups of increasing age, we found subsequently stronger associations between warm weather events and MS-related hospital visits, as well as substantial regional and seasonal heterogeneity wherein the risk of hospital visits was greatest when warmer temperatures may be unexpected, possibly leading to fewer individual-level precautions.

As the global climate changes in response to anthropogenic greenhouse gas emissions, weather and temperature are expected to become increasingly variable. Collectively, our results underscore the importance of ongoing study of the effects of weather and climate anomalies among vulnerable population subgroups, including individuals with MS, in varied geographic settings.

## Supporting information

**S1 STROBE Checklist.**
(DOC)

**S1 Analysis Plan.**
(DOCX)

**S1 Fig. Average case detection rate for MS by state, 2003–2017.** Shading intensity represents the number of MS cases detected per 100,000 person-months of follow-up in the Optum database by state from 2003 to 2017. Patients with MS were identified from the Optum's Clinformatics Data Mart database using a previously validated algorithm, which required at least 3 MS-related inpatient, outpatient, or DMT claims within a 365-day period. MS-related claims were identified using primary diagnostic code 340 from the ICD-9 and G35 from the ICD-10. Prescription claims for DMT were identified as any dispensation for interferon beta-1a-SC, interferon beta-1a-IM, interferon beta-1b-SC, pegylated interferon beta 1b, glatiramer acetate, dimethyl fumarate, fingolimod, siponimod, teriflunomide, cladribine, mitoxantrone, alemtuzumab, ocrelizumab, or natalizumab. Maps were produced using publicly available US Census base maps accessed via the "usmaps" package for R Statistical Software v 4.0.0. DMT, disease-modifying therapy; ICD, International Classification of Diseases; MS, multiple sclerosis.
(TIF)

**S2 Fig. Frequency of MS-related healthcare visits state in the Optum Clinformatics Database, 2003–2017.** Shading intensity represents the percentage of person-months in which at least 1 MS-related outpatient, emergency department, or inpatient visit, respectively, for each state from 2003 to 2017 among the 106,225 patients with MS identified in the Clinformatics

Data Mart Database. MS-related inpatient and outpatient claims were identified using primary diagnostic code 340 from the ICD-9 and G35 from the ICD-10. Maps were produced using publicly available US Census base maps accessed via the "usmaps" package for R Statistical Software v 4.0.0. ICD, International Classification of Diseases; MS, multiple sclerosis.
(TIF)

**S3 Fig. Average monthly temperature and MS-related visits, 2003–2017.** The above figure depicts the predicted probability of MS-related outpatient, emergency department, and inpatient visits at mean monthly temperatures between −10 and 30°C. We used generalized linear models with the binomial family and log link specified. We captured potential nonlinearities in the exposure–response with mean monthly temperatures using natural splines with 3 degrees of freedom. All models were adjusted for continuous age with a natural spline and sex (male or female), and we included a set of fixed effects for calendar year and state. MS, multiple sclerosis.
(TIF)

**S1 Table. Temperature variation (°C) by region and season, 2003–2017.**
(DOCX)

**S2 Table. Anomalously warm weather and MS-related outpatient, emergency department, and inpatient visits, 2003–2017.** MS, multiple sclerosis.
(DOCX)

**S3 Table. Excess MS-related visits and anomalously warm weather, 2003–2007.** MS, multiple sclerosis.
(DOCX)

**S4 Table. Anomalously warm weather and MS-related visits by age category, 2003–2017.** MS, multiple sclerosis.
(DOCX)

**S5 Table. Anomalously warm weather and MS-related visits by season, 2003–2017.** MS, multiple sclerosis.
(DOCX)

**S6 Table. Anomalously warm weather and MS-related visits by US census region, 2003–2017.** MS, multiple sclerosis.
(DOCX)

**S7 Table. Anomalously warm weather and MS-related visits by region and season, 2003–2017.** MS, multiple sclerosis.
(DOCX)

**S8 Table. Anomalously warm weather and MS-related visits by climate zone, 2003–2017.** MS, multiple sclerosis.
(DOCX)

**S9 Table. Difference between average monthly and long-term average temperatures, 2003–2017.**
(DOCX)

**S10 Table. Number of anomalously warm days per month and MS-related visits, 2003–2017.** MS, multiple sclerosis.
(DOCX)

**S11 Table. Alternative thresholds for anomalously warm weather, 2003–2017.**
(DOCX)

**S12 Table. Lagged exposure variable and MS-related visits, 2003–2017.** MS, multiple sclerosis.
(DOCX)

**S13 Table. Negative outcome control: anomalously warm weather and visits unrelated to MS, 2003–2017.** MS, multiple sclerosis.
(DOCX)

## Acknowledgments

Data for this project were accessed using the Stanford Center for Population Health Sciences Data Core, and statistical analyses were performed using the secure Nero Computing Platform. We would like to acknowledge William Law, Neal Soderquist, and the Stanford Research Computing Center for providing computational resources and support that contributed to our study.

## Author Contributions

**Conceptualization:** Holly Elser, Robbie M. Parks, Nuriel Moghavem, Nina Bozinov, Victor W. Henderson, Joan A. Casey.

**Data curation:** Holly Elser, Mathew V. Kiang.

**Formal analysis:** Holly Elser, Robbie M. Parks, Mathew V. Kiang, Joan A. Casey.

**Funding acquisition:** Mathew V. Kiang, Joan A. Casey.

**Investigation:** Holly Elser, Nuriel Moghavem, Nina Bozinov, Victor W. Henderson, David H. Rehkopf, Joan A. Casey.

**Methodology:** Holly Elser, Robbie M. Parks, Mathew V. Kiang, David H. Rehkopf, Joan A. Casey.

**Project administration:** David H. Rehkopf.

**Resources:** David H. Rehkopf.

**Supervision:** David H. Rehkopf, Joan A. Casey.

**Visualization:** Holly Elser.

**Writing – original draft:** Holly Elser.

**Writing – review & editing:** Holly Elser, Robbie M. Parks, Nuriel Moghavem, Mathew V. Kiang, Nina Bozinov, Victor W. Henderson, David H. Rehkopf, Joan A. Casey.

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
