## [Editor Report · Decision Letter 0]

2 Nov 2020

Dear Dr Elser, 

Thank you for submitting your manuscript entitled "Anomalously warm weather and acute care visits in patients with multiple sclerosis: A retrospective study of privately insured individuals in the U.S." for consideration by PLOS Medicine.

Your manuscript has now been evaluated by the PLOS Medicine editorial staff as well as by an academic editor with relevant expertise and I am writing to let you know that we would like to send your submission out for external peer review.

Kind regards,

Artur A. Arikainen,

Associate Editor

PLOS Medicine

---

## [Decision Letter · Decision Letter 1]

30 Nov 2020

Dear Dr. Elser,

Thank you very much for submitting your manuscript "Anomalously warm weather and acute care visits in patients with multiple sclerosis: A retrospective study of privately insured individuals in the U.S." (PMEDICINE-D-20-05226R1) for consideration at PLOS Medicine. 

Your paper was evaluated by a senior editor and discussed among all the editors here. It was also discussed with an academic editor with relevant expertise, and sent to three independent reviewers, including a statistical reviewer (r#2). The reviews are appended at the bottom of this email and any accompanying reviewer attachments can be seen via the link below: 

[LINK]

In light of these reviews, I am afraid that we will not be able to accept the manuscript for publication in the journal in its current form, but we would like to consider a revised version that addresses the reviewers' and editors' comments - please note the academic editor had quite extensive comments that the authors would need to respond to. Obviously we cannot make any decision about publication until we have seen the revised manuscript and your response, and we plan to seek re-review by one or more of the reviewers. 

We expect to receive your revised manuscript by Dec 21 2020 11:59PM. Please email us (plosmedicine@plos.org) if you have any questions or concerns.

We look forward to receiving your revised manuscript. 

Sincerely,

Emma Veitch, PhD

PLOS Medicine

On behalf of Adya Misra, PhD, Senior Editor, 

PLOS Medicine

plosmedicine.org

*Collectively the reviewers raise some concerns around the analyses that are presented, recommend some changes in the statistical analysis approach (including how some data are coded and categorised); they also query whether the main conclusions are plausible given the data. We'd ask that these concerns are thoroughly addressed, which may include moderating some of the conclusions given those points.

*Please structure your abstract using the PLOS Medicine headings (Background, Methods and Findings, Conclusions - "Methods and Findings" is a single subsection). In the last sentence of the Abstract Methods and Findings section, please include a point about any key limitation(s) of the study's methodology.

*Please reformat the citation style into PLOS Medicine's format (should be straight forward if using referencing software) - this should use callouts formatted as sequential numerals in square brackets (not superscript).

*At the moment, it's not clear that the abstract (particularly in the sentence copied below) clearly presents the main findings of the study's analysis. Specifically - "We estimated that at least 1,960 outpatient (95% CI: 1,126–2794), 592 emergency department (95%

CI: 305–878), and 1,260 inpatient (95% CI: 892–1,628) MS-related visits occurred during periods of anomalously warm weather" - does this mean those numbers pertain to "additional" visits corresponding to the increased risk attributed to higher temperature? If so, that should be stated. Simply saying that those numbers of visits occur during the higher temperature periods, doesn't establish that there is a link or association. The authors could also consider including the results of primary analysis for the adjusted risk ratios attributed to higher temperatures. (In the main m/s, given as RR 1.044 (for emergency department visits, with 95%CI 1.025 - 1.063) and RR 1.033 for inpatient visits (95% CI 1.011–1.055)).

*At this stage, we ask that you include a short, non-technical Author Summary of your research to make findings accessible to a wide audience that includes both scientists and non-scientists. The Author Summary should immediately follow the Abstract in your revised manuscript. This text is subject to editorial change and should be distinct from the scientific abstract. Please see our author guidelines for more information: https://journals.plos.org/plosmedicine/s/revising-your-manuscript#loc-author-summary

*Please ensure that the study is reported according to the STROBE guideline (http://www.equator-network.org/reporting-guidelines/strobe/), and include the completed STROBE checklist as Supporting Information. Please add the following statement, or similar, to the Methods: "This study is reported as per the Strengthening the Reporting of Observational Studies in Epidemiology (STROBE) guideline (S1 Checklist)." When completing the checklist, please use section and paragraph numbers, rather than page numbers.

*We'd ask the authors to clarify if the main analytical approach reported here corresponds to one laid out in a prospective protocol or analysis plan. Please state this (either way) early in the Methods section.

Academic editor comments: 

My understanding of the authors' conceptual framework is that "sudden increases in ambient or core body temperature" may lead to exacerbations of MS. My major comments have to do with the implied causality within this framework:

1. Why is this analysis conducted using person-months rather than person-days? These are insurance claim data, so presumably the authors have access to exact dates. They could then revise the exposure variable to be = 1 if the temperature within a time window that is much smaller (e.g., daily temperature, or average daily temperature within a 3-day window, etc.) than the time window used in the present analysis (monthly) exceeds the long-term average by 1.5C. R2's comment about lagged effect is relevant here.

2. Related to the above, why is the exposure dichotomized? Is there reason to expect that a greater number of _days_ of exposure to anomalous weather could produce more exacerbations compared with fewer days of exposure? One could imagine an analysis in which the authors calculate anomalous weather in days -21 to -14, days -13 to -7, days -6 to -1, day 0, days 1-6, days 7-13, days 14-21, etc. (For the post-exposure bins, please see my comment below about placebo exposure.) Alternatively, is there any reason to expect that greater exposure to anomalous weather would lead to more severe exacerbations (e.g. longer hospital stays)? Or, is there reason to expect that larger deviations from average weather in warmer regions/seasons might lead to more severe exacerbations? Some of the examples in the text (e.g., 2003 French heat wave) are true "heat waves" -- when the temperature spikes to 98 in a Chicago summer [+15 deg compared with the monthly average] one would expect to see many more MS exacerbations compared with days when the temperature spikes to 65 in a Boston November [+15 deg compared with the monthly average]).

(The authors have used a number of different thresholds for temperature difference, e.g. +0.5, +1.0, +2.0 deg, but why not model exposure as a categorical variable or using a spline, rather than as binary?)

3. Why is the baseline reference identified as the "long-term average for that month and county over the study period"? If the hypothesized mechanism is that MS exacerbations are due to "sudden increases in ambient or core body temperature", wouldn't a more appropriate referent be something like the average temperature over the prior month?

4. I appreciate that the authors have estimated region- and season-specific models. But on this point I am in agreement with R2 who asks: "Is an excess of 1.5 degrees in winter really heat stress?" It seems reasonable to hypothesize that warm weather anomalies might produce exacerbations during warm weather months (e.g., when the temperature is 100 degrees instead of 95 degrees as expected), but is it reasonable to expect a symmetric effect during cold weather months? (Do hospitals in Boston experience an increase in acute MS care utilization when the temperature hits 70 degrees in November?) I am surprised that the strongest associations were observed in the winter months. Perhaps the manuscript would benefit from additional detail

5. I am in agreement with R2 that the negative control/falsification analysis -- perhaps it would be reasonable to cite either Lipsitch et al. (Epidemiol 2010;21:383-388) or Prasad & Jena (JAMA 2013;309:241-242) -- does not seem to be providing the hoped-for reassurance. In addition to the analysis presented, might the authors consider some type of placebo exposure where they estimate the effect of weather anomalies on MS utilization prior to the exposure period? (Logically, if the hypothesized mechanism is true, the weather anomaly cannot cause MS exacerbations _prior_ to the weather anomaly.)

6. Is it standard in the field to accept all visits flagged with ICD9=340 as being "multiple sclerosis related"? I would like to see a little more specificity here (e.g., "all visits with primary diagnosis ICD9=340 _and_ procedure code = 99.23 ("injection of steroid") (e.g., Laura A et al. Environ Res. 2016;145:68–73). In any case, I would expect the authors to revise the outpatient visits outcome variable along these lines (e.g., either ICD9=340 as a primary _or_ secondary diagnosis _and_ a steroid claim within 7 days -- see Ollendorf DA et al. J Manag Care Pharm 2002;8:469-76; and Chastek BJ et al. J Med Econ 2010;13:618-625).

Minor comments:

7. Was seasonality ("winter", "spring", etc) specific to latitude or altitude? (e.g., December in San Francisco is very different from December in Boston; and December in Gallup, New Mexico is very different from December in Phoenix, Arizona; and temperature spikes of +15 deg are much more manageable in Phoenix, where everyone has air conditioning, compared with in Boston or San Francisco, where many people do not have air conditioning)

8. In the negative control analysis, the authors specify that MS-unrelated visits were "any visit without diagnostic codes 340 or G35"-- are they referring to just the primary diagnosis or also the secondary diagnosis?

9. Was the exposure adjusted for humidity/heat index? If not, why not?

10. Please consider shifting some of the supplemental tables to the primary text. Some of the supplemental tables are appropriately left in the appendix. But some of the supplemental tables address (or hope to address) aspects of the Bradford Hill criteria, so one might hope that these would be shifted to the main text.

Summary:

This is an interesting manuscript, but given the relatively small magnitude effect sizes (and relatively small e-value), I need more convincing that the estimated effect is not simply a byproduct of the sample size.

Comments from the reviewers:

Reviewer #1: 

Review of the original article" Anomalously warm weather and acute care visits in patients with multiple sclerosis: A retrospective study of privately insured individuals in the U.S"

The paper sets out to investigate the effect of heat on acute care visits in patients with multiple sclerosis. The topic is timely is timely and interesting and the investigation of patients suffering from disorders in the central nervous system is needed. The authors have performed sound analyses complemented with relevant secondary and sensitivity analyses. However, some concerns need to be addressed before this study can be considered for publication.

Major comments:

Does the claims data set include date of death? What happened if an individual changed state of residence or changed insurance? The follow-up period should be described in more detail.

Even though the authors divide US into regions to investigate how an increase/decrease in temperature impacts care visits, and the cut-off of 1.5C is sound in a climate change perspective, a country wide study would benefit from adding some information on where this change is on the relative scale. That is, an increase of 1.5 C, may not be the same on the relative scale, even within region, which then would have implications on interpreting the results as one may be comparing a more "extreme" increase to one that is less so. 

Minor comments:

The references 2-6 used to support that morbidity have been attributed to extreme heat previously are not attribution studies per se.

Usually a 30-year window is used in climate studies. What would be the long term county average using 30-year windows instead of using the average over the study period and how would this affect the results?

Other:

Present the RR and 95% CI in a consistent manner. Now not always the case.

Reviewer #2: See attachment

Michael Dewey

Reviewer #3: This is an elegant study of the impact of temperature anomalies on MS-specific healthcare utilization making excellent use of extant big data.

The authors have been diligent in recognizing the inherent limitations in these data and have adopted appropriate methodological and analytical strategies to control for these limitations.

My only criticism, suggestions really, would be to modify the abstract to report the RR as opposed to the excess number of visits. Given the very small increased risk, reporting the excess number of visits overstates the magnitude of the observed effect. You could state in the abstract that the observed effect was increased when a 2.0C threshold was used to address magnitude of the effect.

Overall, this was a well written report of a complex study with a number of different analytical steps. Kudos.

[LINK]

---

## [Decision Letter · Decision Letter 2]

8 Mar 2021

Dear Dr Elser, 

On behalf of my colleagues and the Academic Editor, [AE Name], I am pleased to inform you that we have agreed to publish your manuscript "Anomalously warm weather and acute care visits in patients with multiple sclerosis: A retrospective study of privately insured individuals in the U.S." (PMEDICINE-D-20-05226R2) in PLOS Medicine.

PRESS

Sincerely, 

Dr Raffaella Bosurgi 

Executive Editor 

PLOS Medicine